



# Effect of Inorganic-to-Organic Mass Ratio on the Heterogeneous OH Reaction Rates of Erythritol: Implications for Atmospheric Chemical Stability of 2-Methyltetrols

Rongshuang Xu[1], Hoi Ki Lam[1], Kevin R. Wilson[2], James F. Davies[3], Mijung Song[4], Wentao Li[5], Ying-Lung Steve Tse[5], Man Nin Chan [1,6]

[1]Earth System Science Programme, Faculty of Science, The Chinese University of Hong Kong, Hong Kong, China

[2]Chemical Sciences Division, Lawrence Berkeley National Laboratory, Berkeley, CA, USA

[3]Department of Chemistry, University of California Riverside, Riverside, CA, USA

[4]Department of Earth and Environmental Sciences, Jeonbuk National University, Jeollabuk-do, Republic of Korea

[5]Departemnt of Chemistry, The Chinese University of Hong Kong, Hong Kong, China

[6]The Institute of Environment, Energy and Sustainability, The Chinese University of Hong Kong, Hong Kong, China

Corresponding author: mnchan@cuhk.edu.hk

**Abstract**

2-methyltetrols have been widely chosen as chemical tracers for isoprene-derived secondary organic aerosols. While they are often assumed to be relatively unreactive, a laboratory study reported that pure erythritol particles (an analog of 2-methyltetrols) can be heterogeneously oxidized by gas-phase OH radicals at a significant rate. This might question the efficacy of these compounds as tracers in

aerosol source apportionment studies. Additional uncertainty could raise since organic compounds and inorganic salts are often coexisted in atmospheric particles. To gain more insights into the chemical

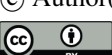



stability of 2-methyltetrols in atmospheric particles, this study investigates the heterogeneous OH oxidation of pure erythritol particles and particles containing erythritol and ammonium sulfate (AS) at different dry inorganic-to-organic mass ratios (IOR) in an aerosol flow tube reactor at a high relative humidity of 85 %. The same reaction products are formed upon heterogenous OH oxidation of

5       erythritol and erythritol-AS particles, suggesting that the reaction pathways are not strongly affected by the presence and amount of AS. On the other hand, the effective OH uptake coefficient, $\gamma_{eff}$, is found to decrease by about a factor of ~20 from $0.45 \pm 0.025$ to $0.02 \pm 0.001$ when the relative abundance of AS increases and the IOR increases from 0.0 to 5.0. One likely explanation is the presence of dissolved ions slows down the reaction rates by decreasing the surface concentration of erythritol and reducing

the frequency of collision between erythritol and gas-phase OH radicals at the particle surface. Hence, the heterogeneous OH reactivity of erythritol and likely 2-methyltetrols in atmospheric particles would be slower than previously thought when the salts are present. Given 2-methyltetrols often coexist with a significant amount of AS in many environments, where ambient IOR can vary from ~1.89 to ~250, our kinetic data would suggest that 2-methyltetrols in atmospheric particles are likely chemically stable

against heterogeneous OH oxidation under humid conditions.

## 1. Introduction

The photochemical oxidation of isoprene is one of major sources of atmospheric secondary organic aerosols (SOA), which can potentially affect the regional and global air quality (Claeys et al., 2004;

Carlton et al., 2009; Wennberg et al., 2018). In many aerosol source apportionment studies, 2-methyltetrols have been used as chemical tracers to quantify the contribution of isoprene-derived SOA to ambient particle organic mass (Kourtchev et al., 2005; Xia and Hopke, 2006; Lewandowski et al., 2007; Zhang et al., 2013; Xu et al., 2014; D'Ambro et al., 2017; He et al., 2018). While 2-methyltetrols are generally considered to be unreactive, this hypothesis has not been thoroughly tested. For instance,

the heterogeneous oxidation of organic compounds present at particle surface by gas-phase oxidants such as hydroxyl (OH) radicals, ozone ($O_3$) and nitrate radicals, has been shown to be efficient in



various laboratory and modeling studies (Rudich et al., 2007; George and Abbatt, 2010; Kroll et al., 2015; Chapleski et al., 2016; Estillore et al., 2016; Huang et al., 2018). Using the erythritol (Table 1) as a surrogate for 2-methyltetrols, Kessler et al. (2010) reported that pure erythritol particles can be heterogeneously oxidized by gas-phase OH radicals at a significant rate with an effective OH uptake

coefficient, $\gamma_{eff}$, of $0.77 \pm 0.1$ and a corresponding chemical lifetime of $\sim 13.8 \pm 1.4$ days at a relative humidity (RH) of 30 %. These results suggest that the abundance of 2-methyltetrols reported in the literature and the amount of isoprene-derived SOA predicted using the chemical tracer method could be underestimated – if heterogeneous oxidation and other atmospheric removal processes (e.g. hydrolysis and aqueous-phase oxidation) have not been properly considered in aerosol source

apportionment studies (Kessler et al., 2010).

To date, the heterogeneous kinetics and chemistry of many pure organic compounds or organic mixtures have been investigated (Zhang et al., 2015; Enami et al., 2016; Socorro et al., 2017; Marshall et al., 2018; Zhao et al., 2019). There remains large uncertainty on how inorganic salts alter the

heterogeneous kinetics and chemistry of organic compounds (McNeill et al., 2007, 2008; Dennis-Smither et al., 2012). A few recent laboratory studies have revealed that the presence of dissolved inorganic ions (e.g. ammonium sulfate, AS) can reduce the heterogeneous OH reactivity of organic compounds in aqueous organic-inorganic particles (e.g. methanesulfonic acid and 3-methylglutaric acid), but does not significantly alter the reaction mechanisms (Mungull et al., 2017; Kwong et al.,

2018a; Lam et al., 2019a). In these studies, only a single inorganic-to-organic mass ratio was chosen to examine the impacts of the salts on the heterogeneous reactivity of organic compounds. However, organic compounds and inorganic salts often coexist with varying concentrations in the atmosphere. For instance, field studies have reported that the mass concentration of 2-methyltetrols ranges from 1-100's ng m$^{-3}$ while that of AS ranges from few to tens µg m$^{-3}$ (Schauer et al., 2002; Lewandowski et

al., 2007; Kleindienst et al., 2010; Budisulistiorini et al., 2013; Xu et al., 2014; Hu et al., 2015). The mass concentration ratio of sulfate-to-2-methyltetrols can vary greatly from ~1.89 to ~250. Since the





impacts of the salts on the heterogeneous reactivity would depend on the relative abundance of organic compounds and inorganic salts, investigations on how the amount of the salts (or the inorganic-to-organic mass ratio) alters heterogeneous reactivity are necessary.

To gain more insights into the chemical transformation and stability of 2-methyltetrols in the atmosphere, experiments were conducted to investigate the heterogeneous OH oxidation of pure erythritol particles and particles containing erythritol and AS in different (water-free) inorganic-to-organic mass ratios (IOR or AS-to-erythritol mass ratio) at 85 % RH using an aerosol flow tube reactor (Table 1). The real-time chemical characterization of the particles before and after OH oxidation was

carried out using a soft atmospheric pressure ionization source (Direct Analysis in Real Time, DART) coupled with a high-resolution mass spectrometer. Erythritol is chosen as a surrogate for investigating the heterogeneous reactivity of 2-methyltetrols (Kessler et al., 2010) while AS is chosen as a common atmospheric inorganic salt. We acknowledge that the IOR investigated in this work (IOR = 0.0–5.0) lies at the low range of IOR observed in atmospheric particles. This would better represent the

environments where the emission and photochemical activities of isoprene are significant. By examining the molecular evolution of pure erythritol particles and erythritol-AS particles during oxidation, we investigate how the presence and concentration of AS affects the heterogeneous OH kinetics and chemistry of erythritol. The results of this work will provide further insights into the chemical stability of 2-methyltetrols in atmospheric particles against heterogeneous OH oxidation.

## 2. Experimental Method

### 2.1 Heterogeneous Oxidation of Erythritol Particles and Erythritol–AS Particles

The heterogeneous OH oxidation of erythritol particles and erythritol–AS particles was investigated using an aerosol flow tube reactor at 20 ℃ and 85 % RH. Experimental details were given elsewhere

(Cheng et al., 2016; Chim et al., 2017a, b; Lam et al., 2019a). Briefly, aqueous droplets were atomized using an atomizer and were directly mixed with ozone, oxygen ($O_2$), dry nitrogen ($N_2$), humidified $N_2$





and hexane before being introduced into the reactor. Inside the reactor, the particles were oxidized by gas-phase OH radicals, which were generated by the photolysis of ozone in the presence of ultraviolet light at 254 nm and water vapor. The gas-phase OH concentration was adjusted by varying the ozone concentration and was determined by measuring the change in the concentration of hexane before and

after OH oxidation using a gas chromatograph coupled with a flame ionization detector (GC-FID) (Smith et al., 2009). The OH exposure, defined as the products of gas-phase OH concentration and particle residence time, was varied from 0.0 to ~$2.29 \times 10^{12}$ molecule cm$^{-3}$ s in all experiments with the particle residence time of 90s.

After leaving the reactor, the ozone and gas-phase species in the particle stream were removed by passing through an annular Carulite catalyst denuder and an activated charcoal denuder, respectively, to allow the sampling of particle-phase products. A portion of the particle stream was then sampled by a scanning mobility particle sizer (SMPS) for particle size measurements. The remaining flow was directed to a heater, where the particles were fully vaporized. In separate experiments, erythritol

particles and erythritol–AS particles were confirmed to be fully vaporized at 300 ºC or above upon heating. The gas-phase species leaving the heater were introduced into an atmospheric pressure ionization region, a narrow open space between the DART ionization source (IonSense: DART SVP), and the inlet orifice of the high-resolution mass spectrometer (Thermo Fisher, Q Exactive Oibritrap) for ionization and detection (Nah et al., 2013; Chan et al., 2013; 2014).

In the ionization region, the electrons (e$^-$) produced by the Penning ionization of metastable He in the DART ionization source were captured by atmospheric $O_2$ molecules to form anionic oxygen ions ($O_2^-$) which then react with gas-phase species (Cody et al., 2005). Nah et al. (2013) have reported that erythritol can be detected as its deprotonated molecular ion, [M−H]$^-$, which can be formed via the

proton abstraction from one of the hydroxyl groups of erythritol by $O_2^-$. As discussed later, carboxylic acids are likely formed upon oxidation and can be detected as [M−H]$^-$ as well (Nah et al., 2013). The





resultant ions were sampled by the high-resolution mass spectrometer and the particle-DART mass spectra were analyzed using the Xcalibur software (Thermo Fisher Scientific).

Control experiments were also carried out to investigate the effects of $O_3$ and UV light on particles:

one in the presence of $O_3$ without the UV light and one in the presence of UV light without $O_3$. There were no significant changes in the DART-particle mass spectra in both control experiments for erythritol and erythritol−AS particles, indicating that the erythritol does not likely react with ozone and is not likely to be photolyzed. Additionally, no erythritol signal was observed when erythritol particles and erythritol−AS particles were removed from the particle stream by filtration using a particle filter

before entering the heater. This suggests the evaporation of erythritol was not significant under our experimental conditions, which agrees with the results reported by Kessler et al. (2010).

### 2.2 Physical State and Mixing Time Scale of Erythritol Particles

The physical state of particles can play a key factor in determining the composition, morphology and

properties (e.g. viscosity) of the particles, which in turn influencing the heterogeneous reactivity (Renbaum and Smith, 2009; Slade and Knopf, 2014; Fan et al., 2015; Marshall et al., 2018; Karadima et al., 2019). Marsh et al. (2017) have measured the hygroscopicity of erythritol particles and found that erythritol particles are spherical droplets over their experimental RH range (60−100 %). Given the hygroscopic data and erythritol particles were always exposed to high RH (i.e. 85 %) and did not pass

through a diffusion dryer in our system, they were likely aqueous droplets before oxidation. At low RH, erythritol particles are known to be viscous (Song et al., 2016; Grayson et al., 2017; Chu et al., 2018). With high particle viscosity, the diffusion of erythritol molecules from the bulk phase to the particle surface for oxidation slows down, and the overall heterogeneous reaction rate can be controlled by the diffusion (Chim et al., 2018; Marshall et al., 2018). To investigate whether particle viscosity

affects the heterogeneous OH reactivity of erythritol particles at 85 % RH in this work, we calculated the characteristic time scale for diffusive mixing time ($\tau_D$) of erythritol within the particle and that for


gas phase OH−erythritol particle collisional timescale ($\tau_{coll}$) (Chim et al., 2018). The $\tau_D$ can be calculated as follow (Abbatt et al., 2012):

$$\tau_D = \frac{D_p{}^2}{4 D_{org}\,\pi^2} \qquad (1)$$

where $D_p$ is the mean surface-weighted diameter of erythritol particles ($D_p$ = 276.1 nm), and $D_{org}$ is the diffusion coefficient of erythritol in the particle and can be estimated using the Stokes–Einstein Equation (Laguerie et al., 1976):

$$D_{org} = \frac{k_b T}{6\pi\eta R_H} \qquad (2)$$

where $D_{org}$ is the diffusion coefficient (m$^2$ s$^{-1}$), $k_b$ is the Boltzmann constant, $T$ is the temperature, $\eta$ is the viscosity ($\eta$ = 1.9× 10$^{-3}$ Pa s at 293 K and 85 % RH) (Song et al., 2016), and $R_H$ is the hydrodynamic radius of an erythritol molecule ($R_H$ = 0.34 ± 0.01 nm) (Chu et al., 2018). Using Eq. 2, $D_{org}$ is calculated to be 3.54 × 10$^{-10}$ m$^2$ s$^{-1}$, and the $\tau_D$ is estimated to be 5.45 × 10$^{-6}$ s at 85 % RH. The collision timescale between erythritol particles and gas-phase OH radicals, $\tau_{coll}$, can be estimated from the collision frequency ($J_{coll}$) of gas-phase OH radicals on the particle surface (Chim et al., 2018):

$$J_{coll} \cong \frac{[OH]\,\overline{c_{OH}}\,A}{4} \qquad (3)$$

where $\overline{c_{OH}}$ is the mean thermal velocity of gas-phase OH radicals, and $A$ is the surface area of the erythritol particle. Using Eq. 3, $\tau_{coll}$ (= 1/$J_{coll}$) is estimated to be about 2.36 × 10$^{-6}$ s at the maximum OH exposure (i.e. highest gas-phase OH concentration). In this study, as the diffusive mixing time scale ($\tau_D$ = 5.45 × 10$^{-6}$ s) and OH-particle collisional timescale ($\tau_{coll}$ = 2.36 × 10$^{-6}$ s) are estimated to be in the same order of magnitude, erythritol could be reasonably assumed to be well-mixed within the particles under our experimental conditions.

## 2.3 Physical State and Mixing Time Scale of Erythritol–AS Particles

Organic compounds and inorganic salts usually coexist in atmospheric particles. In response to atmospheric conditions (e.g. RH and temperature) and particle composition, these mixed particles can undergo phase transition (deliquescence and crystallization) and phase separation (e.g. solid-liquid and



liquid–liquid phase separation) (Braban and Abbatt, 2004; Song et al., 2012a, b; You et al., 2013, 2014; Veghte et al., 2014; Karadima et al., 2019). Laboratory studies have reported that no phase separation was observed for organic–inorganic particles when organic compounds with O/C ratio larger than 0.8. Although the physical state of erythritol–AS particles has not been experimentally measured, erythritol–AS particles are likely aqueous droplets at high RH (i.e. 85 % RH) and exists as a single aqueous phase prior to oxidation as erythritol has an O/C ratio of one (Table 1). To our best knowledge, the viscosity of erythritol–AS particles with different IOR has not been reported in the literature. However, for the purposes of this work we will assume that erythritol is well mixed within all erythritol–AS particles prior to oxidation under our experimental conditions.

## 3. Results and Discussions

### 3.1 Particle-DART Mass Spectra of Erythritol and Erythritol–AS Particles

The particle-DART mass spectra of erythritol and erythritol–AS particles with different IORs before and after OH oxidation are shown in Figure 1. For erythritol particles, before oxidation, only one major peak of the deprotonated molecular ion of erythritol ($C_4H_9O_4^-$) at *m/z* 111 is observed, together with some small background peaks. After oxidation, at the highest OH exposure, unreacted erythritol has the largest signal and accounts for 27.8 % of the total ion signal. Two major product peaks evolve. The $C_4$ functionalization products ($C_4H_8O_5$) and $C_3$ fragmentation products ($C_3H_6O_4$) contribute 23.4 % and 17.9 % of the total ion signal, respectively. Some small product peaks ($C_4H_7O_4^-$, $C_4H_5O_4^-$, $C_4H_3O_4^-$ and $C_3H_5O_3^-$) are also detected.

The particle-DART mass spectra of erythritol–AS particles are very similar to that of erythritol particles, except for an inorganic sulfate peak. Before oxidation, there are two major peaks at *m/z* 97 and *m/z* 111, corresponding to the bisulfate ion ($HSO_4^-$) and the deprotonated molecular ion of erythritol ($C_4H_9O_4^-$), respectively. After oxidation, the $C_4$ functionalization products ($C_4H_8O_5$) and $C_3$ fragmentation products ($C_3H_6O_4$) are the two major products together with some minor product peaks.



The $HSO_4^-$ is likely originated from AS. For the erythritol–AS particles, AS dissociates into the ammonium and sulfate ions. Upon heating and evaporation, the particles becomes acidified by the evaporative loss of ammonia into gas phase and the dissolved sulfate ions can be detected as $HSO_4^-$ via direct ionization (Hajslova et al., 2011; Lam et al., 2019a, b; Kwong et al., 2018a, b). The intensity of $HSO_4^-$ before and after oxidation (Figure S1, *supplementary material*) had no significant change, which is consistent with the argument made by George and Abbatt (2010) that the surface reaction between dissolved ions and gas-phase OH radicals is not efficient. As same reaction products are observed for both erythritol particles and erythritol–AS particles with different IORs, these results suggest that the heterogenous OH reaction mechanisms of erythritol do not significantly affect by the presence and amount of AS.

As shown in Figure 1, the deprotonated molecular ion of erythritol is the dominant peak in the mass spectra before oxidation, suggesting that the thermal decomposition of erythritol might not be significant. These results are consistent with the literature (Nah et al., 2013). Nah et al. (2013) have also showed that the deprotonated molecular ions are the dominant ions for carboxylic acids in their particle-DART analysis. Taken together, these results indicate that the effect of thermal decomposition on the observed products may be insignificant. However, we would like to note that some possible reaction products (e.g. organic peroxides and oligomers could be formed from reactions between peroxy radicals) may thermally decompose (Stark et al., 2017). We do not rule out the formation of these products upon OH oxidation of erythritol, but there was no indication of fragment ions expected from the thermal decomposition of these products in the particle-DART mass spectra. Further investigation on the formation of these products during heterogeneous OH oxidation of organic compounds is desirable. In the following, the heterogeneous OH kinetics and chemistry of the erythritol and erythritol–AS particles is examined based on the particle-DART mass spectra measured at different extent of oxidation.



### 3.2 Oxidation Kinetics of Erythritol and Erythritol–AS Particles

Figure 2a shows the normalized decay of erythritol in erythritol particles and erythritol–AS particles with different IOR as a function of OH exposure. For all these particles, the OH-initiated decay of erythritol exhibits an exponential trend and can be fit with an exponential function:

$$ln\frac{I}{I_0} = -k\,[OH]\cdot t \tag{4}$$

where $I$ is the signal intensity of erythritol at a given OH exposure, $I_0$ is the signal intensity before oxidation, $k$ is the effective second-order heterogeneous OH rate constant, and $[OH]\cdot t$ is the OH exposure. It can be seen that the rate of reaction decreases with deceasing amount of erythritol or increasing amount of AS (Table 1). When the IOR increases from 0.0 to 5.0, the $k$ decreases from $5.39 \pm 0.12 \times 10^{-13}$ to $1.56 \pm 0.04 \times 10^{-13}$ cm$^3$ molecule$^{-1}$ s$^{-1}$. Further, the effective OH uptake coefficient, $\gamma_{eff}$, defined as the fraction of OH collisions with erythritol molecule that result in a reaction, can be computed (Kessler et al., 2010; Davies and Wilson, 2015),

$$\gamma_{eff} = \frac{2}{3}\frac{D_p\,\rho\,mfs\,N_A k}{M\,\overline{c_{OH}}} \tag{5}$$

where $D_p$ is the mean surface-weighted particle diameter before OH oxidation, $\rho$ is the particle density before oxidation, $N_A$ is the Avogadro's number, $mfs$ is the mass fraction of erythritol, $M$ is the molecular weight of erythritol, and $\overline{c_{OH}}$ is the average speed of gas-phase OH radicals. For erythritol particles, before oxidation, the mean surface-weighted particle diameter was 276.1 nm. The density of erythritol particles is estimated to be 1.173 g cm$^{-3}$, using the volume additivity rule with the density of water and erythritol (1.451 g cm$^{-3}$) and particle composition (i.e. mass fraction of solute, $mfs$). The $mfs$ is derived from the hygroscopicity data reported by Marsh et al. (2017) and is reported to be $0.47 \pm 0.02$ at 85 % RH, which agrees well with model simulations ($mfs = 0.482$) using the Aerosol Inorganic–Organic Mixtures Functional groups Activity Coefficients (AIOMFAC) model (Zuend et al., 2008; Zuend et al., 2011). For erythritol–AS particles, the particle diameters were measured to be 278.2–281.5 nm before oxidation. The $mfs$ of erythritol in erythritol–AS particles are obtained from the model simulation using AIOMFAC to be 0.280, 0.210, 0.061 at IOR = 0.5, 1.0 and 5.0, respectively (Table





1). Based on the composition of erythritol–AS particles (i.e. *mfs*), the particle density was also estimated using the volume additivity rule with the density of water, erythritol and AS (1.77 g cm$^{-3}$). Using Eq. 5, the $\gamma_{eff}$ is calculated to be 0.45 ± 0.025 for erythritol particles (i.e. IOR = 0). For erythritol–AS particles, the $\gamma_{eff}$ is calculated to be 0.20 ± 0.010, 0.12 ± 0.006, 0.02 ± 0.001 at IOR = 0.5, 1.0 and

5.0, respectively. Figure 2b shows that the heterogeneous reactivity of erythritol toward gas-phase OH radicals as a function of IOR at 85 % RH. The $\gamma_{eff}$ is found to decrease from 0.45 ± 0.025 to 0.02 ± 0.001 when the IOR increases from 0.0 to 5.0.

These results agree with the literature that the addition of AS decreases the overall rate of

heterogeneous OH oxidation with organic compounds (Mungull et al., 2017, Kwong et al., 2018a; Lam et al., 2019a). We carried out molecular dynamics (MD) simulations to gain a better insight into the effect of dissolved ions on the heterogeneous OH reactivity of erythritol. The details of MD simulation are given in the *supplementary material*. Previous simulation results suggest excess kinetic energy that an impinging gas molecule may carry will dissipate in a few ps after collision (Vieceli et al., 2005; Li

et al., 2019). The difference between this very short timescale and the experimental timescale of the reaction on ms timescale indicates that the reaction is not likely initiated by the direct collision between the gas-phase OH radical and the erythritol molecule present at the particle surface. Our simulation also shows that it is not easy for the gas-phase OH radical to collide near an erythritol molecule at first impact with or without salt. With salt, the probability for the gas-phase OH radical to collide near an

erythritol molecule becomes even lower because of lower concentrations as shown in Figure S3, suggesting the reaction via direct impact is unlikely.

Some gas-phase OH radicals would be absorbed by the particle after collision, and the reaction would require an absorbed OH radical and an erythritol to meet many times by diffusion before the reaction

could happen. To shed light on how likely absorbed OH radical meets erythritol within the droplet, the probability densities of the distance between the centers of mass (COMs) of the OH radical and the





closest erythritol molecule in presence and in absence of salt were calculated. As shown in Figure S2, in the presence of (hydroscopic) salt, the erythritol–AS particle contains more water and the concentrations of erythritol and adsorbed OH radical are smaller, making the average distance between the OH radical and its nearest erythritol longer relative to pure erythritol particle. The longer average

distance would slow down the reaction rate, which is consistent with the decreased reaction rate in the experiment in the presence of salt.

Another possibility for this lower heterogeneous reactivity might due to the change in surface-bulk partitioning behavior of organic compound in the presence of AS which could potentially alter the

surface concentration of organic reactant. Previous studies have found that the addition of AS could resulted in a pronounced increase/decrease in particle surface tension compared to that of organic/water particle, indicating a salting in/out effect (Ekström et al., 2009; Zhang and Carloni, 2012; Boyer and Dutcher, 2017; Fan et al., 2019). These effects might result in smaller/larger surface concentration of organic than that in the bulk and further affect the overall reactivity. For instance, the smaller surface

concentration of reactive species could ultimately lower the reactive collision probability between the organic compound and OH radicals, slowing down the overall heterogeneous oxidation rate. While our model simulations could not provide the density profile of erythritol molecules within particles, future investigations are desirable to better understand how the concentration of organic molecule at the particle surface would affect the heterogeneous reactivity in the presence and absence of the salt.

Based on the kinetic data, we also estimate the chemical lifetime of erythritol against heterogeneous OH oxidation, $\tau$ under different particle composition (i.e. IOR) at 85 % RH (Kroll et al., 2015):

$$\tau = \frac{[\text{Erythritol}]}{d[\text{Erythritol}]/dt} = \frac{1}{k[OH]} \tag{6}$$

As shown in Table 1, assuming a 24 h averaged OH concentration of $1.5 \times 10^6$ molecule cm$^{-3}$ (Mao et

al., 2009), the $\tau$ of pure erythritol particles is estimated to be $14.7 \pm 0.33$ days. The timescales are





slightly longer than those of other important particle removal processes, such as dry and wet deposition (~ 5–12 days) with the similar particle size (200 nm) (Kanakidou et al., 2005). A similar result has been reported in the literature. Kessler et al. (2010) have investigated the heterogeneous OH oxidation of pure erythritol particles at a lower RH (30 % RH) and reported a chemical lifetime of about $13.8 \pm 1.4$ days. These results suggest that the variation of RH does not significantly alter the rate of OH reaction with erythritol. On the other hand, the reaction rates depend on the concentration of erythritol and AS. The chemical lifetime increases from $14.7 \pm 0.33$ days to $49.5 \pm 1.43$ days when the IOR increases from 0.0 to 5.0 (Table 1). These indicate that erythritol become more chemically stable against OH oxidation when the salt is present. We acknowledge that the highest IOR investigated in this work (IOR = 5.0) lies at the low range of IOR reported for the 2-methyltetrols in atmospheric particles. The results of this work might provide insights into how 2-methyltetrols chemically age through heterogeneous OH oxidation in the environments where the emission and photochemical activities of isoprene are significant. We also note that 2-methyltetrols is often mixed with a large amount of AS and the ambient IOR can be as large as ~250. Since the heterogeneous reactivity decreases with increasing IOR, large ambient IOR values may suggest that the heterogeneous OH reactivity of 2-methyltetrols in atmospheric particles would be much slower than previously predicted based on experiments with pure organic particles. It would be reasonable to assume that 2-methyltetrols are likely chemically stable against heterogeneous OH oxidation over their atmospheric timescales.

### 3.3 Proposed Reaction Mechanisms

As shown in Figure 1, same reaction products are observed for both erythritol and erythritol–AS particles, suggesting the presence and the amount of AS does not significantly affect the formation pathways of major reaction products. We tentatively propose the same reaction pathways for OH reaction with erythritol in the absence and presence of AS based on particle-phase reactions previously reported in the literature (Bethel et al., 2003; Kessler et al., 2010; George and Abbatt, 2010; Kroll et al., 2015). As shown in Scheme 1, OH oxidation with erythritol can be initiated by the hydrogen



abstraction from the two C−H bonds (Path A and Path B) and two O−H bonds (Path C and Path D). A

variety of functionalization (Section 3.3.1) and fragmentation products (Section 3.3.2) can be formed

when gas-phase OH radicals attack different reaction sites.

### 3.3.1 Functionalization Products

Scheme 1 shows a variety of functionalization products can be possibly formed during oxidation. The

formation of functionalization products is likely originated from the hydrogen abstraction occurred at

the C−H bonds (Scheme 1: Path A and Path B). This is because when the hydrogen abstraction occurs

on the O−H groups (Scheme 1: Path C and Path D), the resultant alkoxy radicals tend to decompose

into smaller fragmentation products.

$C_4H_8O_5$ As shown in Scheme 1 (Path A), the major $C_4$ functionalization product ($C_4H_8O_5$), as shown

in Figure 1, can be formed when the hydrogen abstraction occurs at the secondary carbon site. At the

first oxidation step, an alkyl radical is formed after hydrogen abstraction by OH radicals and quickly

reacts with an oxygen molecule to form a peroxy radical. The self-reactions of two peroxy radicals can

produce the $C_4$ carboxylic acid ($C_4H_8O_5$) via well-known particle-phase reactions such as Russell and

Bennett and Summers reactions (Russell, 1957; Bennett and Summers, 1973). While a carboxylic acid

group is formed during oxidation, the effective saturation vapor pressure, $C^*$ of the $C_4$ carboxylic acid

is estimated to be 0.195 μg m$^{-3}$ using the saturation vapor pressure predicted by EVAPORATION

(Compernolle et al., 2011). Given its low volatility, it likely remains in particle phase upon production.

$C_4H_8O_4$ A small peak has been observed for another $C_4$ functionalization products ($C_4H_8O_4$) (Figure

1). The formation of these products could be originated from the unimolecular HO$_2$ elimination of

hydroxyperoxy radicals (Bothe et al., 1983; Bethel et al., 2003; Kessler et al., 2010). For instance,

when the hydrogen abstraction occurs at the secondary carbon site (Scheme 1: Path A), a

hydroxyperoxy radical can undergo the unimolecular HO$_2$ elimination process to form a $C_4$



hydroxyaldehyde ($C_4H_8O_4$). Moreover, when the hydrogen abstraction occurs at the tertiary carbon site (Scheme 1: Path B), a $C_4$ hydroxyketone ($C_4H_8O_4$) can be formed via the same process. This unimolecular $HO_2$ elimination process is expected to be kinetically favorable for polyols (Bothe et al., 1978a, b). This is because the hydroxyl group adjacent to the peroxy group can stabilize the resultant

carbonyl group by forming strong intramolecular hydrogen bond, thus enhancing the unimolecular $HO_2$ elimination rate (Bothe et al., 1978a, b; Cheng et al., 2016). As show in Figure 1, the ion signal intensity of these functionalization products is only less than 5 % of the total ion signal. Although the ionization efficiencies of these products were not corrected in this study, our earlier study found that the ionization efficiency of ketone products is higher than alcohol products with same carbon number

during the DART ionization processes (Chan et al., 2014). Thus the low abundance of the ketone/aldehyde products might be better attributed to the high volatilities of these functionalization products. When the unimolecular $HO_2$ elimination occurs, a hydroxyl group is being converted into a carbonyl group. This increases the volatilities of reaction products compared to their parent compounds. For instance, the $C^*$ of the $C_4$ hydroxyketone (Scheme 1: Pathway B) and the $C_4$ hydroxyaldehyde

(Scheme 1: Pathway A) is estimated to be $3.01 \times 10^2$ µg m$^{-3}$ and $6.95 \times 10^2$ µg m$^{-3}$, respectively. The volatilities of these two products are predicted to be about 1–2 orders of magnitude larger than that of pure erythritol ($C^* = 5.71$ µg m$^{-3}$).

### 3.3.2 Fragmentation Products

**$C_3H_6O_4$** Fragmentation products can be generated from the decomposition of alkoxy radicals during oxidation. The major $C_3$ fragmentation product is likely originated from the hydrogen abstraction at the tertiary carbon site (Scheme 1: Pathway B). The alkoxy radicals resulted from peroxy-peroxy reactions can fragment to form a $C_2$ hydroxyketone ($C_2H_4O_2$) and a $C_3$ carboxylic acid ($C_3H_6O_4$). The $C_2$ hydroxyketone is volatile ($C^* = 3.75 \times 10^7$ µg m$^{-3}$) and likely partitions back to the gas phase. For

the $C_3$ carboxylic acid, although a carbon atom is lost, the formation of the carboxylic acid functional group lowers its volatility ($C^* = 5.64$ µg m$^{-3}$). Thus, it is expected to be nonvolatile and likely remains



in the particle phase. Additionally, the structure–activity relationship (SAR) model developed for the decomposition of alkoxy radicals suggests that the formation of the larger fragmentation product (i.e. $C_3$ carboxylic acid) is more kinetically favorable than that of the smaller fragmentation product (i.e. $C_2$ hydroxyketone) upon decomposition as the formation rate coefficient ($k_{SAR}$) of the $C_3$ fragmentation

product is $7.13 \times 10^{12}$ s$^{-1}$, which is about three order of magnitude higher than that of the $C_2$ fragmentation product ($3.40 \times 10^9$ s$^{-1}$) (Peeters et al., 2004; Vereecken et al., 2009).

During oxidation, a number of fragmentation products can be possibly formed. However, the ion signals of these fragmentation products that remained in the particle phase are very small or not

detected in the particle-DART mass spectra (Figure 1). This might attribute to the volatilization of these fragmentation products. For instance, when the hydrogen abstraction occurs at the secondary carbon site (Scheme 1: Path A), the decomposition of the alkoxy radical yields a formic acid (HCOOH) ($C^*$ of $1.90 \times 10^7$ µg m$^{-3}$). Alkoxy radicals can be directly formed when the hydrogen abstraction occurs at the two OH-groups (Scheme 1: Path C and Path D). The decomposition of resultant alkoxy

radicals can yield volatile fragmentation products such as HCOOH, a $C_3$ hydroxyaldehyde ($C_3H_6O_3$, $C^* = 1.75 \times 10^5$ µg m$^{-3}$) and a $C_2$ hydroxyketone ($C_2H_4O_2$, $C^* = 3.75 \times 10^7$ µg m$^{-3}$). These products likely partition back to the gas phase due to their high volatilities.

Overall, the formation of two major products detected in the particle-DART mass spectra (Figure 1) is

likely originated from two distinct reaction sites. The major functionalization products ($C_4$ carboxylic acid) are likely formed when the hydrogen abstraction occurs at the secondary carbon site followed by functionalization processes (Scheme 1: Path A), while the major fragmentation product ($C_3$ carboxylic acid) are likely formed from the decomposition of an alkoxy radical formed at the tertiary carbon site (Scheme 1: Path B). Further investigations on the DART ionization efficiency and detection of both

particle-phase and gas-phase products are highly desirable to better understand the reaction mechanisms and assess the relative importance between functionalization, fragmentation and



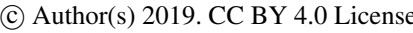

volatilization processes in governing the composition of erythritol particles during heterogeneous OH oxidation.

## 4. Conclusions and Atmospheric Implications

5 To date, there remains considerable uncertainty in how inorganic salts alter the heterogeneous reactivity of organic compounds, which ultimately governs the chemical lifetime of organic compounds in the atmosphere. Here, we investigated the effects of AS on the kinetics, products and mechanisms upon heterogeneous OH oxidation of erythritol at 85 % RH at different (dry) mass ratios of erythritol and AS. Particle-DART mass spectra obtained for both erythritol and erythritol–AS

10 particles showed the same reaction products, suggesting that formation pathways of major reaction products do not significantly affect by the presence and amount of AS. On the other hand, the heterogeneous reactivity of erythritol toward gas-phase OH radicals could be slower in erythritol–AS particles compared to pure erythritol particles, depending on the concentration of erythritol and AS. This could be explained by that the colliding probability between OH radical and erythritol in the

15 particle and at the particle surface become lower in the presence of salts, resulting in a smaller overall reaction rate. Overall, our results provide evidence that inorganic salts likely alter the heterogeneous reactivity of organic compounds with gas-phase OH radicals rather than the reaction mechanisms. Further, our kinetic data suggest that given the ambient concentration of 2-methyltetrols and AS reported in field measurements, 2-methyletrols in the atmospheric particles are likely chemical stable

20 against heterogeneous OH oxidation under humid conditions.

### Data availability

The underlying research data are available upon request from the corresponding author (mnchan@cuhk.edu.hk).

### Author contributions





Rongshuang Xu and Man Nin Chan designed and ran the experiments. Rongshuang Xu, Hoi Ki Lam, and Man Nin Chan prepared the manuscript. All co-authors provided comments and suggestions to the manuscript.

**Competing interests**

The authors declare that they have no conflict of interest.

**Acknowledgements**

Rongshuang Xu, Hoi Ki Lam and Man Nin Chan are supported by the Hong Kong Research Grants
Council (HKRGC) Project ID: 2130626 (Ref 14300118). Kevin R. Wilson is supported by the Department of Energy, Office of Science, Office of Basic Energy Sciences, Chemical Sciences, Geosciences, and Biosciences Division under Contract No. DE-AC02-05CH11231.

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





**Table 1.** Chemical structure, properties, effective rate constant and OH uptake coefficient of pure erythritol particles and erythritol−AS particles with different IORs at 85 % RH.

| Compounds | Erythritol | | | |
|---|---|---|---|---|
| Structural Formula |  | | | |
| Molecular Formula | $C_4H_{10}O_4$ | | | |
| Molecular Weight (g mol$^{-1}$) | 122.12 | | | |
| Particle Composition | | | | |
| Inorganic-to-Organic Mass Ratio (IOR) | 0.0 (Pure) | 0.5 | 1.0 | 5.0 |
| Sulfate-to-Organic Mass Ratio | 0.0 (Pure) | 0.36 | 0.72 | 3.64 |
| Mass Fraction of Erythritol (%) | 47.3 | 28.0 | 19.9 | 6.1 |
| Mass Fraction of AS (%) | 0.0 | 14.0 | 19.9 | 30.5 |
| Mass Fraction of Water (%) [a] | 52.7 | 58.0 | 60.2 | 63.4 |
| Particle Density (g cm$^{-3}$) | 1.172 ± 0.010 | 1.175 ± 0.005 | 1.177 ± 0.004 | 1.182 ± 0.001 |
| Effective Saturation Vapor Pressure of Erythritol, C* (μg m$^{-3}$) [b] | 0.686 | 0.792 | 0.921 | 1.60 |
| Effective Heterogeneous OH Oxidation Rate Constant, $k$ (×10$^{-13}$ cm$^3$ molecule$^{-1}$ s$^{-1}$) | 5.38 ± 0.12 | 4.00 ± 0.04 | 3.26 ± 0.05 | 1.56 ± 0.04 |
| Effective OH Uptake Coefficient, $\gamma_{eff}$ | 0.45 ± 0.025 | 0.20 ± 0.010 | 0.12 ± 0.006 | 0.02 ± 0.001 |
| Chemical lifetime (days) [c] | 14.3 ± 0.33 | 19.3 ± 0.22 | 23.6 ± 0.37 | 49.5 ± 1.43 |

[a] The amount of water is predicted using the aerosol thermodynamic model at 85 % RH before oxidation

5    [b] Effective saturated vapor pressure of erythritol predicted before oxidation

[c] 24-h averaged OH concentration of $1.5 \times 10^6$ molecules cm$^{-3}$

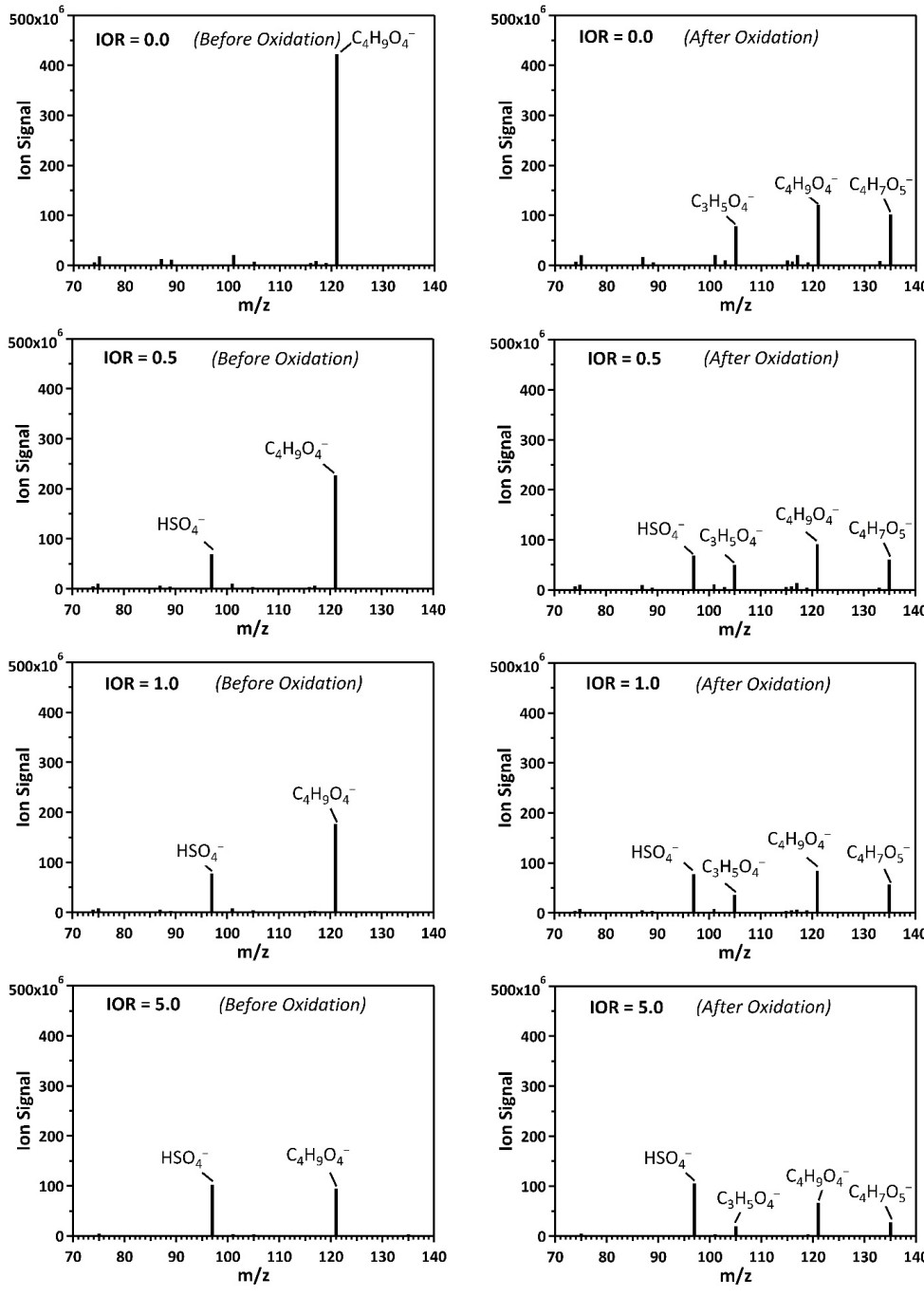

**Figure 1.** Particle-DART mass spectra for erythritol particles and erythritol−AS particles at different

IORs before and after oxidation (at the highest OH exposure of ∼ $2.29 \times 10^{12}$ molecule $cm^{-3}$ s).



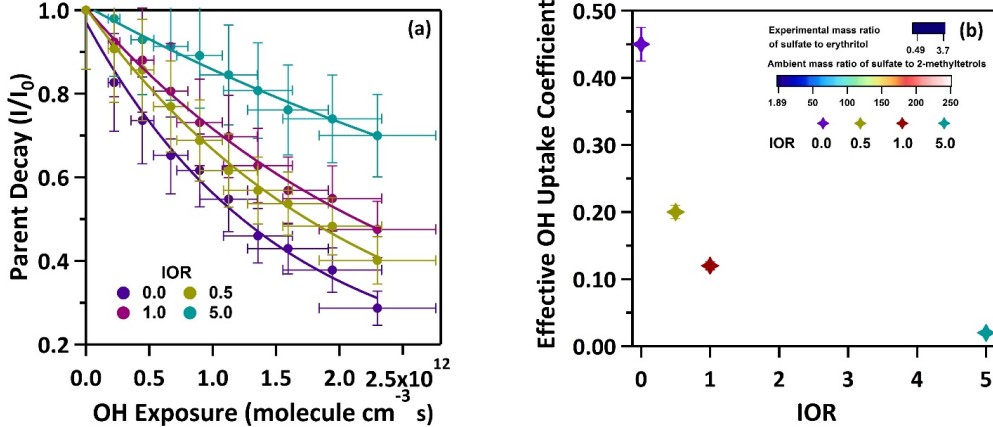

**Figure 2.** (a) The normalized decay of erythritol as a function of OH exposure during the heterogeneous OH oxidation of erythritol particles and erythritol-AS particles with different IORs. (b) The effective OH uptake coefficient, $\gamma_{\text{eff}}$.





**Scheme 1.** Reaction mechanisms tentatively proposed for the heterogeneous OH oxidation of erythritol.