# Peer review of "Effect of Inorganic-to-Organic Mass Ratio on the Heterogeneous OH Reaction"

_Atmospheric Chemistry and Physics, 2019_

## Referee Comment (RC1) · Anonymous Referee #1 · 30 Dec 2019

This is the review of the manuscript entitled "Effect of Inorganic-to-Organic Mass Ratio on the Heterogeneous OH Reaction Rates of Erythritol: Implications for Atmospheric Chemical Stability of 2-Methyltetrols" by Xu et al.

This study examines the photochemical stability of pure erythritol particles and particles containing erythritol and ammonium sulfate (AS) against oxidation by OH radicals. Erythritol serves as a surrogate of 2-methyltetrols, a common compound of isoprene-derived secondary organic aerosols. Reactive uptake of OH and subsequent degradation of erythritol in the particulate phase were determined as a function of inorganicto-organic mass ratio (IOR). This was achieved using an aerosol flow reactor coupled to a soft atmospheric pressure ionization source (Direct Analysis in Real Time, DART) attached to a high-resolution mass spectrometer. It is found that the reactive uptake coefficient of OH decreases as the amount of AS increases. Furthermore, the results suggest that the reaction products due to OH oxidation are not significantly affected by the presence of AS. Since in the ambient methyltetrols are associated with AS, this study concludes that the chemical lifetime of methyltetrols is prolonged and may render methyltetrols stable against OH oxidation under humid conditions.

The topic of this study fits well within the scope of ACP. I do not have major comments on this work, mostly minor ones and technical in nature. However, I suggest to carefully proofread the manuscript to improve the English language.

P. 9, top paragraph: Just a comment to better understand the origin of HSO4-. As the particles are heated and vaporize (ammonia is gone), the SO42- can abstract a hydrogen from the organic molecule? This leads to the detection of HSO4-? Since HSO4- signal does not change significantly between non-oxidized and the oxidized case, reactivity of OH with SO42- or HSO4- (and maybe NH4+) is not significant? A bit of rewording and better integration of previous studies would make this section easier to understand.

Derivation of the OH uptake coefficient necessitates the particle diameter. It appears that a polydisperse aerosol was applied. It would be helpful to plot the aerosol size distribution or give the corresponding distribution parameters. Did the authors measure the aerosol distribution before and after oxidation to assess potential volatilization of the particles upon oxidation?

How is the uncertainty of presented uptake values derived? Likely the width of the aerosol size distribution and the uncertainty in AIOMFC derived mfs values contribute to the overall uncertainty?

Addition of the MD simulation is a neat feature. However, its validity depends strongly

on established parameters, applied fields, etc. It would have been nice to show that the simulated system or another test case behaves as expected. Here, things may get more complicated since NH4+ and SO42- ions will be differently distributed within the particles? This could not be addressed as stated in text but could be discussed looking at previous studies (e.g., Tobias and Jungwirth groups).

P. 12., l. 10-13: Please be more specific. Do you expect salting in or salting out for the particle systems you investigated? Will it change for the different IOR?

Figure 2b: The legends using color bars are a bit confusing and maybe, I misunderstand those. Does the color coding of the experimental data has a relationship with the color coding of the ambient sulfate to 2-methyltetrol ratio? No ambient data is plotted and thus one wonders the meaning of that legend. The text already stated that the ambient ratio is much higher than the one probed in experiments. Same with the sulfate to erythritol ratio. There is no change in the color bar and the data are not plotted as a function of this ratio. This could be done by using a 3D plot. Also, the legends are not described in figure caption.

Suggested technical corrections:

p. 1, l. 26: . . .inorganic salts often coexist in atmospheric. . ..

p. 9, l. 1: The HSO4- likely originated. . ..

p. 9, l. 5: . . .before and after oxidation (. . .) showed no significant change. . .

---

## Referee Comment (RC2) · Anonymous Referee #2 · 13 Jan 2020

Xu et al. investigated the OH-initiated heterogeneous oxidation of erythritol particles and particles containing erythritol and ammonium sulfate (AS). Erythritol was used as a surrogate of 2-methyltetrols, one of the most important isoprene-derived SOA products. SOA chemical composition was retrieved using a soft atmospheric pressure ionization source (DART) coupled to an Orbitrap. The reactivity of erythritol was characterized as a function of inorganic-to-organic mass ratio. While the study is well constrained and clearly presents some interesting results, some aspects are not considered within the discussion (see comments below).

Page 2, lines 23-24: The authors stated that the reactivity of 2-methyltetrols has not been tested before. That's not fully correct. Hu et al. (Jimenez's group) investigated the aging of ambient isoprene-derived SOA and found that IEPOX-SOA (mainly producing 2-MT) are fairly unreactive. This study should be cited by the authors and further discuss.

Page 3, lines 3-5: The authors should be careful here and do not overstate the impact of OH reactivity. A chemical lifetime of 2 weeks cannot be really classified as a "significant" reaction/loss.

Page 4, lines 26:The authors should provide more information regarding the experimental conditions: - What was the gas-phase concentration of erythritol? Is it possible that larger IOR lead to higher degassing (e.g., salting-out effect?) - Can the presence of gaseous erythritol decrease the concentration of OH radicals and further impact the heterogeneous reactivity? - Please provide the size, surface area and mass of particles for each condition.

Page 6, dart section: Please provide more information. Was it real-time evaporation or were the particles collected onto a filter prior vaporization?

Page 9: Recent studies have shown that isoprene-derived SOA, especially when formed in the presence of acidic aerosols are highly viscous, further impacting heterogeneous processes (e.e., Surratt's group, Ault's group, Thornton's group). While the assumption that erythritol is well mixed is likely correct, the authors cannot ignore these recent studies and the last paragraph (i.e., atmospheric implications) should mention the impact of the phase and viscosity on the heterogeneous reactivity of isoprene-derived SOA products. In other words, the rate constant/lifetimes proposed in this study are likely an upper limit suggesting that the OH oxidation of 2-methyltetrols is negligible.

Page 9, lines 11-14: Did the particles shrunk? Did the authors estimate a carbon closure?

**ACPD**
Page 10, lines 9-10: That doesn't mean that thermal decomposition is not occurring for other types of compounds, e.g., carboxylic acid or other oxygenated species. As the authors did not report any concentrations it is not difficult to make such a statement; i.e., the observed compounds can be only a fraction of the quantity formed. In addition, the measured compounds can fragment into small ions:

---

## Author Comment (AC1) · 23 Feb 2020

*This study examines the photochemical stability of pure erythritol particles and particles containing erythritol and ammonium sulfate (AS) against oxidation by OH radicals. Erythritol serves as a surrogate of 2-methyltetrols, a common compound of isoprene derived secondary organic aerosols. Reactive uptake of OH and subsequent degradation of erythritol in the particulate phase were determined as a function of inorganic-to-organic mass ratio (IOR). This was achieved using an aerosol flow reactor coupled to a soft atmospheric pressure ionization source (Direct Analysis in Real Time, DART) attached to a high-resolution mass spectrometer. It is found that the reactive uptake coefficient of OH decreases as the amount of AS increases. Furthermore, the results suggest that the reaction products due to OH oxidation are not significantly affected by the presence of AS. Since in the ambient methyltetrols are associated with AS, this study concludes that the chemical lifetime of methyltetrols is prolonged and may render methyltetrols stable against OH oxidation under humid conditions. The topic of this study fits well within the scope of ACP. I do not have major comments on this work, mostly minor ones and technical in nature. However, I suggest to carefully proofread the manuscript to improve the English language.*

**We would like to sincerely thank the reviewer for his/her thoughtful comments. The referee's comments are below in italics followed by our responses.**

**Comment #1:**
*P. 9, top paragraph: Just a comment to better understand the origin of $HSO_4^-$. As the particles are heated and vaporize (ammonia is gone), the $SO_4^{2-}$ can abstract a hydrogen from the organic molecule? This leads to the detection of $HSO_4^-$? Since $HSO_4^-$ signal does not change significantly between non-oxidized and the oxidized case, reactivity of OH with $SO_4^{2-}$ or $HSO_4^-$ (and maybe $NH_4^+$) is not significant? A bit of rewording and better integration of previous studies would make this section easier to understand.*

**Author Response:**
Thanks for the comments. We would argue that sulfate ion ($SO_4^{2-}$) is not an oxidant and cannot abstract a hydrogen atom from an organic molecule to form bisulfate ion ($HSO_4^-$). As discussed in the manuscript and in the literature, $HSO_4^-$ likely originates from ammonium sulfate (AS). In our experiments, before introduced to the ionization region, erythritol–AS particles were fully vaporized under high temperature inside the particle heater and may thus thermally decompose into gas-phase ammonia ($NH_3$) and sulfuric acid ($H_2SO_4$) (Drewnick et al., 2015), which can be detected as $HSO_4^-$ via direct ionization (Hajslova et al., 2011; Lam et al., 2019a, b; Kwong et al., 2018a, b).

George and Abbatt (2010) have revealed that dry $(NH_4)_2SO_4$ surface is highly unreactive toward gas-phase OH radicals. Furthermore, the surface reaction between dissolved $SO_4^{2-}$ and gas-phase OH radicals is not efficient (Cooper and Abbatt, 1996;

Anastasio and Newberg, 2007). We have added this information in the revised manuscript.

Page 9, Line 12: "The $HSO_4^-$ likely originated from AS. Before introduced to the ionization region, erythritol–AS particles were fully vaporized under high temperature and may thus thermally decompose into gas-phase $NH_3$ and $H_2SO_4$ (Drewnick et al., 2015), which can be detected as $HSO_4^-$ via direct ionization (Hajslova et al., 2011; Lam et al., 2019a, b; Kwong et al., 2018a, b). The intensity of $HSO_4^-$ before and after oxidation (Figure S1, *supplementary material*) showed no significant change, which is consistent with the argument in previous studies (Cooper and abbatt, 1996; Anastasio and Newberg, 2007) that the surface reaction between dissolved sulfate ions and gas-phase OH radicals is not efficient."

**Comment #2:**
*Derivation of the OH uptake coefficient necessitates the particle diameter. It appears that a polydisperse aerosol was applied. It would be helpful to plot the aerosol size distribution or give the corresponding distribution parameters. Did the authors measure the aerosol distribution before and after oxidation to assess potential volatilization of the particles upon oxidation?*

**Author Response:**
Thanks for the comments. In our experiments, polydisperse particles were applied and the geometric standard deviation was 1.2–1.3. We acknowledge that the span of polydisperse particles could have effects on the determination of effective OH uptake coefficient, $\gamma_{eff}$. Further study which measures the $\gamma_{eff}$ for both size-selected monodisperse and polydisperse particles is desired to better investigate the effect of particle size distribution on $\gamma_{eff}$ calculation. As the spread of particle size in this work is small, we postulate that it would not significantly affect the determination of $\gamma_{eff}$.

We have observed slight changes in particle size upon oxidation for both erythritol and erythritol–AS particles (please see the **Figure S2** below). The surface-weighted mean diameter changes from 276.1 nm to 255.8 nm (~ 7.3 %) for erythritol particles and decreases from 278.2 nm to 262.1 nm (~ 5.8 %), from 280.5 nm to 266.7 nm (~ 4.9 %), from 281.2 nm to 274.0 nm (~ 2.6 %) for erythritol–AS particles at IOR = 0.5, 1.0 and 5.0, respectively. The decrease in particle size could be explained by the formation and volatilization of some reaction products proposed in the reaction scheme (i.e. $C_4$ hydroxyketone and hydroxyaldehyde ($C_4H_8O_4$), $C_2$ hydroxyketone ($C_2H_4O_2$) and $C_3$ hydroxyaldehyde ($C_3H_6O_3$)).

[Figure]

**Figure S2**. The change in surface-weighted mean diameter as a function of OH exposure for erythritol particles and erythritol-AS particles with different IORs.

*Supporting material*, we have added **Figure S2** in the *supporting material* to illustrate the change in particle diameter for erythritol particles and erythritol–AS particles upon oxidation.

Page 10, Line 21: "Further, the initial effective OH uptake coefficient, $\gamma_{eff}$, defined as the fraction of OH collisions with erythritol molecule that result in a reaction, can be computed (Kessler et al., 2010; Davies and Wilson, 2015),"

Page 11, Line 2: "For erythritol particles, the initial mean surface-weighted particle diameter was 276.1 nm and decreased to 255.8 nm after oxidation (~ 7.3 %)."

Page 11, Line 9: "For erythritol–AS particles, the particle diameters were measured to be 278.2–281.5 nm before oxidation (Table 1). Slight decreases in particle diameter (~ 5.8 %, ~ 4.9 %, ~2.6 % at IOR = 0.5, 1.0 and 5.0, respectively) were also observed (*Figure S2, supplementary material*)."

Page 11, Line 19: "The $\gamma_{eff}$ is found to decrease from 0.45 ± 0.025 to 0.02 ± 0.001 when the IOR increases from 0.0 to 5.0. We acknowledge that the span of polydisperse particles could have effects on the determination of $\gamma_{eff}$. Further study which measures the $\gamma_{eff}$ for size selected monodisperse and polydisperse particles is desired to better investigate the effect of particle size distribution on $\gamma_{eff}$ calculation."

**Comment #3:**
*How is the uncertainty of presented uptake values derived? Likely the width of the aerosol size distribution and the uncertainty in AIOMFC derived mfs values contribute to the overall uncertainty?*

**Author Response:**
The uncertainty of effective OH uptake coefficient, $\gamma_{eff}$ is calculated according to the error propagation rule:

$$\sigma_\gamma = \gamma_{eff} * \sqrt{\left(\frac{\sigma_k}{k}\right)^2 + \left(\frac{\sigma_{D0}}{D_0}\right)^2 + \left(\frac{\sigma_\rho}{\rho}\right)^2 + \left(\frac{\sigma_{mfs}}{mfs}\right)^2} \quad (Eqn.1)$$

where $\gamma_{eff}$ is the effective OH uptake coefficient, $\sigma_\gamma$ is the uncertainty of effective OH uptake coefficient, $k$ is the measured effective heterogeneous OH rate constant, $\sigma_k$ is the uncertainty of effective heterogeneous OH rate constant, $D_0$ is the mean surface-weighted diameter, $\sigma_{D0}$ is the uncertainty of the mean surface-weighted particle diameter (± 0.5 % uncertainty), $mfs$ is the mass fraction of solute, $\sigma_{mfs}$ is the uncertainty of mass fraction of solute (± 0.02 for erythritol particles (Marsh et al., 2017), ± 5 % $mfs$ for erythritol–AS particles predicted by AIOMFAC), $\rho$ is the estimated particle density based on the volume additivity rule, $\sigma_\rho$ is the uncertainty of particle density (determined based on following equation, Eqn.2)

$$\sigma_\rho = \rho^2 \frac{IOR*\rho_{o*}\rho_w + \rho_w*\rho_{AS} - (IOR+1)*\rho_o*\rho_{AS}}{\rho_o*\rho_w*\rho_{AS}} * \sigma_{mfs} \quad (Eqn.2)$$

where $\rho_w$ is the water density (1.0 g cm$^{-3}$), $\rho_o$ is the erythritol density (1.451 g cm$^{-3}$), $\rho_{AS}$ is the density of AS (1.77 g cm$^{-3}$), $IOR$ is the inorganic-to-organic ratio. From the calculations, the uncertainty in particle diameter has minor effect on $\sigma_v$ (less than 1 % at all IORs), while the uncertainty in *mfs* contributes about ~ 90 % at all IORs when determining $\sigma_v$.

**Comment #4:**
*Addition of the MD simulation is a neat feature. However, its validity depends strongly on established parameters, applied fields, etc. It would have been nice to show that the simulated system or another test case behaves as expected. Here, things may get more* complicated *since NH$_4^+$ and SO$_4^{2-}$ ions will be differently distributed within the particles? This could not be addressed as stated in text but could be discussed looking at previous studies (e.g., Tobias and Jungwirth groups).*

**Author Response:**
Thanks for the comment. While the previous studies such as that by Tobias and Jungwirth (2001) suggest that the propensity of a species for the air-water interface can depend strongly on the MD force fields used, our explanation is based on two general physical observations that do not depend strongly on the model parameters. Our first observation is that the probability for an impinging OH radical to collide near an erythritol molecule decreases as the concentration of erythritol decreases when there is more water due to the hygroscopicity of a salt. A lower concentration of erythritol leading to a smaller collision probability should not be model-specific. The second observation is that the average distance between an OH radical and an erythritol molecule increases as their concentrations decrease because of more water. This again is a physical argument that does not depend strongly on the models used.

**Comment #5:**
*P. 12., l. 10-13: Please be more specific. Do you expect salting in or salting out for the particle systems you investigated? Will it change for the different IOR?*

**Author Response:**
To our best knowledge, the surface-bulk-partitioning behavior of erythritol when mixed with ammonium sulfate (AS) has not been experimentally measured. Ekström et al. (2009) have measured the surface tension using a FTÅ 125 tensiometer and have reported that when AS was added into 2-methylerythritol (with chemical structures similar to erythritol), the surface tension, σ was found to increase compared to that of 2-methylerythritol. For instance, for the system with 17 wt % of AS and 0.05 M of 2-methylerythritol, the surface tension was ~ 72.6 mN m$^{-1}$, which is larger than that for 2-methylerythritol/water system (σ (0.05 M) = ~69.7 mN m$^{-1}$). In their study, the surface tension increased from ~ 50.3 mN m$^{-1}$ to 72.6 mN m$^{-1}$ when IOR increased from ~0.8 to ~25.0, suggesting a salting in effect.

| Composition | Surface tension (mN m$^{-1}$) | Molar concentration of 2-methylerythritol (M) |
|---|---|---|
| 2-methylerythritol | −14.3 c(M) + 70.4 | 0.02 − 1.87 |
| 2-methylerythritol + (17 % wt/wt) AS | −15.1 c(M) + 73.4 | 0.05 − 1.53[a] |

ᵃ the range of molar concentration of erythritol in our work is from 0.59 M at IOR=5.0 to 4.5 M at IOR=0.0. And the corresponding IOR for 2-methylerythritol/AS/water mixture ranges from ~0.8 to ~25.0.

On the other hand, Riva et al. (2019) have recently observed an interfacial tension depression using a biphasic microfluidic platform when AS is mixed with 2-methyltetrols (1.55 M of AS and 0.37 M of 2-methyltetrols (IOR = ~ 4.1)). This suggests a salting out effect. Based on these two results, the salt effect on the surface-bulk-partitioning behavior of 2-methyltetrols and likely erythritol remains unclear. Future investigations are needed to measure the surface tension for erythritol and erythritol–AS systems at different IORs in order to better understand the effect of salt on surface-bulk-partitioning behavior of erythritol within the particle and overall heterogeneous reactivity. We have added the following information in the revised manuscript.

Page 13, Line 4: "To our best knowledge, the surface-bulk-partitioning behavior of erythritol molecules in the presence of AS has not been experimentally measured. Ekström et al. (2009) have measured the surface tension using a FTÅ 125 tensiometer and have reported that when AS was mixed with 2-methylerythritol (with chemical structures similar to erythritol), the surface tension, σ was found to increase compared to that of 2-methylerythritol. For instance, for the system with 17 wt % of AS and 0.05 M of 2-methylerythritol, the surface tension was ~ 72.6 mN m$^{-1}$, which is larger than that for 2-methylerythritol/water system (σ (0.05 M) = ~69.7 mN m$^{-1}$). In their study, the surface tension increased from ~ 50.3 mN m$^{-1}$ to 72.6 mN m$^{-1}$ when IOR increased from ~0.8 to ~25.0, suggesting a salting in effect. On the other hand, Riva et al. (2019) have recently observed an interfacial tension depression using a biphasic microfluidic platform when AS was mixed with 2-methyltetrols (1.55 M of AS and 0.37 M of 2-methyltetrols (IOR = ~ 4.1)), suggesting a salting out effect. Based on these two results, the salt effect on the surface-bulk-partitioning behavior of 2-methyltetrols and likely erythritol remains unclear. Future investigations which can well represent the distribution of erythritol molecules at the particle surface are desirable to better understand how the presence of salts would alter the surface concentration of organic molecule and ultimately affect its heterogeneous reactivity."

**Comment #6:**
*Figure 2b: The legends using color bars are a bit confusing and maybe, I misunderstand those. Does the color coding of the experimental data has a relationship with the color coding of the ambient sulfate to 2-methyltetrol ratio? No ambient data is plotted and thus one wonders the meaning of that legend. The text already stated that the ambient ratio is much higher than the one probed in experiments. Same with the sulfate to erythritol ratio. There is no change in the color bar and the data are not plotted as a function of this ratio. This could be done by using a 3D plot. Also, the legends are not described in figure caption.*

**Author Response:**
We are sorry for the confusion. We have modified the Figure 2 in the revised manuscript. The color scale represents the range of sulfate to erythritol mass ratio (0 – 3.7) at different IORs in this work, much smaller than that for ambient mass ratio of sulfate to 2-methyltetrols reported in field studies (~1.89 − ~250). Legends are described in the figure caption in the revised manuscript.

Page 36, Line 1: "

[Figure]

**Figure 2.** (a) The normalized decay of erythritol as a function of OH exposure during the heterogeneous OH oxidation of erythritol particles and erythritol-AS particles with different IORs. (b) The effective OH uptake coefficient, $\gamma_{eff}$. The data points represent $\gamma_{eff}$ value at different IORs. The color scale represents the range of sulfate to erythritol mass ratio $(0 - 3.7)$ at different IORs in this work, much smaller than that for ambient mass ratio of sulfate to 2-methyltetrols reported in field studies $(\sim1.89 - \sim250)$."

**Comment #7:**
*p. 1, l. 26: …inorganic salts often coexist in atmospheric…*

**Author Response:**
We have changed this sentence in the revised manuscript.

Page 1, Line 25: "Additional uncertainty could raise since organic compounds and inorganic salts often coexist in atmospheric particles."

**Comment #8:**
*p. 9, l. 1: The $HSO_4^-$ likely originated…*

**Author Response:**
We have revised this sentence.

Page 9, Line 12: "The $HSO_4^-$ likely originated from AS."

**Comment #9:**
*p. 9, l. 5: …before and after oxidation (…) showed no significant change…*

**Author Response:**
We have revised the sentence.

Page 9, Line 15: "The intensity of $HSO_4^-$ before and after oxidation (Figure S1, *supplementary material*) showed no significant change,"

**References**:

Anastasio, C. and Newberg, J. T.: Sources and sinks of hydroxyl radical in sea-salt particles, J. Geophys. Res.: Atmos., 112, doi:10.1029/2006jd008061, 2007.

Bethel, H. L., Atkinson, R. and Arey, J.: Hydroxycarbonyl products of the reactions of selected diols with the OH radical, J. Phys. Chem. A, 107, 6200–6205, doi:10.1021/jp027693l, 2003.

Blanksby, S. J. and Ellison, G. B.: Bond Dissociation Energies of Organic Molecules, ChemInform, 34, doi:10.1002/chin.200324299, 2003.

Block, E., Dane, A. J., Thomas, S., Cody, R. B.: Applications of Direct Analysis in Real Time Mass Spectrometry (DART-MS) in Allium Chemistry. 2-Propenesulfenic and 2-Propenesulfinic Acids, Diallyl Trisulfane S-Oxide, and Other Reactive Sulfur Compounds from Crushed Garlic and Other Alliums, J. Agric. Food Chem., 58, 4617−4625, 2010.

Cooper, P. L. and Abbatt, J. P. D.: Heterogeneous Interactions of OH and $HO_2$ Radicals with Surfaces Characteristic of Atmospheric Particulate Matter, J. Phys. Chem., 100, 2249–2254, doi:10.1021/jp952142z, 1996.

Davies, J. F. and Wilson, K. R.: Nanoscale interfacial gradients formed by the reactive uptake of OH radicals onto viscous aerosol surfaces, Chem. Sci., 6, 7020–7027, doi:10.1039/c5sc02326b, 2015.

Drewnick, F., Diesch, J.-M., Faber, P. and Borrmann, S.: Aerosol mass spectrometry: particle–vaporizer interactions and their consequences for the measurements, Atmos. Meas. Tech., 8, 3811–3830, doi:10.5194/amt-8-3811-2015, 2015.

Ekström, S., Nozière, B., and Hansson, H.: The cloud condensation nuclei (CCN) properties of 2-methyltetrols and $C_3$-$C_6$ polyols from osmolality and surface tension measurements. Atmos. Chem. Phys., 9, 973–980. doi:10.5194/acp-9-973-2009, 2009.

Hajslova, J., Cajka, T., and Vaclavik, L.: Challenging applications offered by direct analysis in real time (DART) in food-quality and safety analysis, TrAC-Trend Anal. Chem., 30, 204–218, https://doi.org/10.1016/j.trac.2010.11.001, 2011.

Kessler, S. H., Smith, J. D., Che, D. L., Worsnop, D. R., Wilson, K. R. and Kroll, J. H.: Chemical sinks of organic aerosol: kinetics and products of the heterogeneous

oxidation of erythritol and levoglucosan, Environ. Sci. Technol., 44, 7005–7010, doi:10.1021/es101465m, 2010.

Kwong, K. C., Chim, M. M., Hoffmann, E. H., Tilgner, A., Herrmann, H., Davies, J. F., Wilson, K. R. and Chan, M. N.: Chemical transformation of methanesulfonic acid and sodium methanesulfonate through heterogeneous OH oxidation, ACS Earth Space Chem., 2, 895–903, doi:10.1021/acsearthspacechem.8b00072, 2018a.

Kwong, K. C., Chim, M. M., Davies, J. F., Wilson, K. R. and Chan, M. N.: Importance of sulfate radical anion formation and chemistry in heterogeneous OH oxidation of sodium methyl sulfate, the smallest organosulfate, Atmos. Chem. Phys., 18, 2809–2820, doi:10.5194/acp-18-2809-2018, 2018b.

Lam, H. K., Shum, S. M., Davies, J. F., Song, M., Zuend, A., and Chan, M. N.: Effects of inorganic salts on the heterogeneous OH oxidation of organic compounds: insights from methylglutaric acid–ammonium sulfate, Atmos. Chem. Phys., 19, 9581–9593, https://doi.org/10.5194/acp-19-9581-2019, 2019a.

Marsh, A., Miles, R. E. H., Rovelli, G., Cowling, A. G., Nandy, L., Dutcher, C. S. and Reid, J. P.: Influence of organic compound functionality on aerosol hygroscopicity: dicarboxylic acids, alkyl-substituents, sugars and amino acids, Atmos. Chem. Phys., 17, 5583–5599, doi:10.5194/acp-17-5583-2017, 2017.

Riva, M., Chen, Y., Zhang, Y., Lei, Z., Olson, N. E., Boyer, H. C., Narayan, S., Yee, L. D., Green, H. S., Cui, T., Zhang, Z., Baumann, K., Fort, M., Edgerton, E., Budisulistiorini, S. H., Rose, C. A., Ribeiro, I. O., Oliveira, R. L. E., Santos, E. O. D., Machado, C. M. D., Szopa, S., Zhao, Y., Alves, E. G., Sá, S. S. D., Hu, W., Knipping, E. M., Shaw, S. L., Junior, S. D., Souza, R. A. F. D., Palm, B. B., Jimenez, J.-L., Glasius, M., Goldstein, A. H., Pye, H. O. T., Gold, A., Turpin, B. J., Vizuete, W., Martin, S. T., Thornton, J. A., Dutcher, C. S., Ault, A. P. and Surratt, J. D.: Increasing Isoprene Epoxydiol-to-Inorganic Sulfate Aerosol Ratio Results in Extensive Conversion of Inorganic Sulfate to Organosulfur Forms: Implications for Aerosol Physicochemical Properties, Environ. Sci. Technol., 53, 8682–8694, doi:10.1021/acs.est.9b01019, 2019.

Jungwirth, P. and Tobias, D. J.: Molecular Structure of Salt Solutions: A New View of the Interface with Implications for Heterogeneous Atmospheric Chemistry, J. Phys. Chem. B, 105, 10468–10472, doi:10.1021/jp012750g, 2001.

---

## Author Comment (AC2) · 23 Feb 2020

*Xu et al. investigated the OH-initiated heterogeneous oxidation of erythritol particles and particles containing erythritol and ammonium sulfate (AS). Erythritol was used as a surrogate of 2-methyltetrols, one of the most important isoprene-derived SOA products. SOA chemical composition was retrieved using a soft atmospheric pressure ionization source (DART) coupled to an Orbitrap. The reactivity of erythritol was characterized as a function of inorganic-to-organic mass ratio. While the study is well constrained and clearly presents some interesting results, some aspects are not considered within the discussion (see comments below).*

**We would like to sincerely thank the reviewer for his/her thoughtful comments. The referee's comments are below in italics followed by our responses.**

**Comment #1:**
*Page 2, lines 23-24: The authors stated that the reactivity of 2-methyltetrols has not been tested before. That's not fully correct. Hu et al. (Jimenez's group) investigated the aging of ambient isoprene-derived SOA and found that IEPOX-SOA (mainly producing 2-MT) are fairly unreactive. This study should be cited by the authors and further discuss.*

**Author Response:**
We would like to thank the reviewer for bringing up this paper by Hu et al. (2016). This work investigated the heterogenous reactivity of ambient IEPOX-SOA towards OH radical and reported reaction kinetics ($4.0 \pm 2.0 \times 10^{-13}$ cm$^3$ molecule$^{-1}$ s$^{-1}$ and $3.9 \pm 1.8 \times 10^{-13}$ cm$^3$ molecule$^{-1}$ s$^{-1}$ for IEPOX-SOA collected in SE US and Amazon, respectively) based on the decay of C$_5$H$_6$O$^+$ ion (a tracer ion for IEPOX-SOA in ambient particles) in their AMS measurements. Based on these data, they calculated on average a more than 2-week ($19 \pm 9$ days) atmospheric lifetime of IEPOX-SOA against OH radical oxidation based on rate constant of $4.0 \pm 2.0 \times 10^{-13}$ cm$^3$ molecule$^{-1}$ s$^{-1}$ and averaged ambient OH concentration of $1.5 \times 10^6$ molecule cm$^{-3}$. We would like to acknowledge that these measured values are for ambient IEPOX-SOA and not for pure 2-methyltetrols. To our best knowledge, the heterogeneous OH reactivity of pure 2-methyltetrols particles has been investigated.

We agree with the reviewer that 2-methyltetrols are important components of IEPOX-SOA in the atmosphere. For instance, 2-methyltetrols can account for 10 % – 20 % of IEPOX-SOA in an experimental study (Surratt et al., 2010) and ~ 24 % of ambient IEPOX-SOA in field studies in rural Alabama, southeastern US (Isaacman et al., 2014). As suggested by Hu et al. (2016), the rates derived in their study could be considered as a lower limit for individual molecular components of IEPOX-SOA (e.g. 2-methyltetrols), as it may take two or more OH reactions for their AMS spectrum to no

longer resemble that of IEPOX-SOA. We have added this information in the revised manuscript.

Page 3, Line 5: "Hu et al. (2016) have investigated the heterogenous reactivity of ambient IEPOX-SOA (consisting of 2-methyltetrols, C5-alkene triols, organosulfate, etc.) towards gas-phase OH radicals in SE US and Amazon and reported the reaction kinetics for IEPOX-SOA based on the decay of $C_5H_6O^+$ ion (a tracer ion for IEPOX-SOA in ambient particles) in their AMS measurements. They calculated on average a more than 2-week (19 ± 9 days) atmospheric lifetime of IEPOX-SOA against heterogeneous OH oxidation based on rate constant of $4.0 ± 2.0 × 10^{-13}$ cm$^3$ molecule$^{-1}$ s$^{-1}$ and averaged ambient OH concentration of $1.5 × 10^6$ molecule cm$^{-3}$. They also suggest that the observed rates may consider as a lower limit for individual molecular components of IEPOX-SOA (e.g. 2-methyltetrols) because it may take two or more OH reactions to make their AMS spectrum distinguishable from that of IEPOX-SOA after oxidation (Hu et al., 2016)."

**Comment #2:**
*Page 3, lines 3-5: The authors should be careful here and do not overstate the impact of OH reactivity. A chemical lifetime of 2 weeks cannot be really classified as a "significant" reaction/loss.*

**Author Response:**
We agree with the reviewer's comment. We have revised this sentence in the manuscript.

Page 3, Line 3: "Kessler et al. (2010) reported that heterogeneous oxidation of pure erythritol particles by gas-phase OH radicals with an effective OH uptake coefficient, $\gamma_{eff}$, of $0.77 ± 0.1$ and a corresponding chemical lifetime of $\sim 13.8 ± 1.4$ days at a relative humidity (RH) of 30 %."

**Comment #3:**
*Page 4, lines 26: The authors should provide more information regarding the experimental conditions: - What was the gas-phase concentration of erythritol? Is it possible that larger IOR lead to higher degassing (e.g., salting-out effect?) - Can the presence of gaseous erythritol decrease the concentration of OH radicals and further impact the heterogeneous reactivity? - Please provide the size, surface area and mass of particles for each condition.*

**Author Response:**
Thanks for the comments. We have mentioned in the manuscript that control experiments had been carried out to investigate the volatilization of erythritol under our experimental conditions. The mass spectra were measured when erythritol particles and erythritol−AS particles were removed from the particle stream by filtration using a particle filter before entering the heater for DART analysis. No obvious peak was observed in mass spectra, suggesting that that there is insignificant amount of erythritol present in the gas phase under our experimental conditions. We would also like to mention that the effective saturation vapor pressure, $C^*$ of erythritol before oxidation for pure erythritol and erythritol−AS systems is estimated based on the framework by Zuend and Seinfeld (2012), using the saturation vapor pressure predicted by

EVAPORATION (Compernolle et al., 2011) and the activity coefficient derived from AIOMFAC. The $C*$ of erythritol is estimated to be 0.686 µg m$^{-3}$, 0.792 µg m$^{-3}$, 0.921 µg m$^{-3}$, 1.60 µg m$^{-3}$ at IOR = 0.0, 0.5, 1.0, 5.0, respectively (**Table 1** in the revised manuscript) and ~ 99 % of erythritol would be expected to remain in particle phase in all experiments based on gas-particle partitioning model (Zuend and Seinfeld, 2012) with the particle mass loading of about 500 µg m$^{-3}$ in our experiments. These results suggest that the volatilization and gas-phase reactivity of erythritol would not be significant in our work. We also acknowledge that the effects of the salt on the gas-particle partitioning of erythritol (in term of C*) have been considered by calculating the activity coefficients of erythritols using AIOMFAC at different IORs.

The surface-weighted mean diameter of erythritol and erythritol–AS particles at all IORs have been given in **Table 1**. The mass loading under all experiments was around 500 µg m$^{-3}$, and the surface area was ~ $9 \times 10^9$ nm² cm$^{-3}$.

**Comment #4:**
*Page 6, dart section: Please provide more information. Was it real-time evaporation or were the particles collected onto a filter prior vaporization?*

**Author Response:**
Thanks for the comment. In our experiments, it was a real-time evaporation of the particles. The particle stream was directed into a stainless-steel tube, where the particles were fully vaporized in real time. The resultant gas-phase species were then delivered into an ionization region. The erythritol particles and erythritol–AS particles were confirmed to be fully vaporized at 300 ℃ before introduced to the ionization region in separate experiments, thus yielding a mass spectrum representative of the entire particle (i.e., bulk composition). We have added this information in the manuscript.

Page 5, Line 22: "The remaining flow was directed to a stainless-steel tube heater, where the particles were fully vaporized in real time. The resultant gas-phase species were then delivered into an ionization region. The erythritol particles and erythritol–AS particles were confirmed to be fully vaporized at 300 ℃ before introduced to the ionization region in separate experiments, thus yielding a mass spectrum representative of the entire particle (i.e., bulk composition)."

**Comment #5:**
*Page 9: Recent studies have shown that isoprene-derived SOA, especially when formed in the presence of acidic aerosols are highly viscous, further impacting heterogeneous processes (e.e., Surratt's group, Ault's group, Thornton's group). While the assumption that erythritol is well mixed is likely correct, the authors cannot ignore these recent studies and the last paragraph (i.e., atmospheric implications) should mention the impact of the phase and viscosity on the heterogeneous reactivity of isoprene derived SOA products. In other words, the rate constant/lifetimes proposed in this study are likely an upper limit suggesting that the OH oxidation of 2-methyltetrols is negligible.*

**Author Response:**
We agree with the reviewer's comment that the particle phase, morphology and viscosity can significantly affect the heterogeneous reactivity of isoprene derived SOA and their individual components (e.g. 2-methyltetrols). When the isoprene derived SOA

are highly viscous, this slows down the overall rate of reaction due to diffusion. In this study, the rate constants and lifetimes measured for well mixed erythritol particles and erythritol–AS particles at a high RH may consider as an upper limit. The OH oxidation of 2-methyltetrols in ambient particles could be slower than our reported values, depending on the formation pathways and composition of isoprene derived SOA and atmospheric conditions. We have added the following information in the conclusions.

Page 18, Line 22: "Recent studies have shown that the phase state and viscosity of the particles depending on the particle composition and environmental factors can significantly affect the diffusivity of organic molecules, water molecules and oxidants such as gas-phase OH radicals, which in turn the overall oxidation rate and formation of products (Chan et al., 2014; Slade and Knopf, 2014; Chim et al., 2017a; Marshall et al., 2016; 2018). Isoprene-derived SOA, especially when formed in the presence of acidic sulfate particles have been reported to be highly viscous (Shrivastava et al., 2017; Olson et al., 2019; Zhang et al., 2019). For instance, Riva et al. (2019) have shown that a viscous IEPOX-SOA coating was likely formed in the presence of acidic sulfate seed particles. The diffusion of organic molecules (e.g. 2-methyltetrols) from the bulk to the particle surface could slow down, lowering the overall heterogeneous reactivity.

To date, the effects of the complex interplay between particle phase, morphology and viscosity on the heterogeneous reactivity remains largely unexplored. We would like to acknowledge that in this study the rate constants and lifetimes measured for well mixed erythritol particles and erythritol–AS particles at a high RH may consider as an upper limit. The OH oxidation of 2-methyltetrols in ambient particles could be slower than our reported values, depending on the formation pathways and composition of IEPOX-SOA and atmospheric conditions (e.g. RH and temperature). All these results suggest that a single kinetic parameter may not be well described for the heterogeneous OH oxidation erythritols and 2-methyltetrols in atmosphere since the rates can vary significantly, depending on the particle composition, phase and morphology and environmental factors."

**Comment #6:**
*Page 9, lines 11-14: Did the particles shrunk? Did the authors estimate a carbon closure?*

**Author Response:**
As discussed in our response for **reviewer 1**, **comment #7**, we have observed slight changes in particle size upon oxidation for both erythritol and erythritol–AS particles (please see the **figure** below). The surface-weighted mean diameter changes from 276.1 nm to 255.8 nm (~ 7.3 %) for erythritol particles and decreases from 278.2 nm to 262.1 nm (~ 5.8 %), from 280.5 nm to 266.7 nm (~ 4.9 %), from 281.2 nm to 274.0 nm (~ 2.6 %) for erythritol–AS particles at IOR = 0.5, 1.0 and 5.0, respectively. The decrease in particle size could be explained by the formation and volatilization of some reaction products (i.e. $C_4$ hydroxyketone and hydroxyaldehyde ($C_4H_8O_4$), $C_2$ hydroxyketone ($C_2H_4O_2$) and $C_3$ hydroxyaldehyde ($C_3H_6O_3$)). We have added this information in the revised manuscript.

[Figure]

**Figure S2**. The change in surface-weighted mean diameter as a function of OH exposure for erythritol particles and erythritol-AS particles with different IORs.

*Supporting material*, we have added **Figure S2** in the *supporting material* to illustrate the change in particle diameter for erythritol particles and erythritol–AS particles upon oxidation.

Determining a carbon closure is a very good suggestion. However, we cannot estimate a carbon closure with a few reasons. One reason is that we may not be able to detect all reaction products with the DART-MS technique. Second, we cannot quantify the concentration of particle-phase products because the ionization efficiencies of the products are not known. At last, gas-phase products have not been measured in our experiments. However, we agree with the reviewer a carbon closure study should be considered and carried out in future study.

**Comment #7:**
*Page 10, lines 9-10: That doesn't mean that thermal decomposition is not occurring for other types of compounds, e.g., carboxylic acid or other oxygenated species. As the authors did not report any concentrations it is not difficult to make such a statement; i.e., the observed compounds can be only a fraction of the quantity formed. In addition, the measured compounds can fragment into small ions: < m/z 70. It is also unclear why the authors selected such a narrow mass range. To fully investigate the potential fragmentation the authors could have extended the mass range: i.e., 50-400. Please clarify.*

**Author Response:**
Thanks for the comments. DART ion source is considered as a less energetic "soft" ionization technique, which applies less excess internal energy to the target molecule, resulting in minimal dissociation and yielding intact ions with minimal fragmentation. Previous studies (Cody et al., 2005; Nah et al., 2013) have shown that the dominant ions for their investigated organic compounds (M), including alkanes, alkenes, carboxylic acids, esters, and alcohols, in negative-ion mass spectra are observed in the form of $[M-H]^-$, or $[M]^-$. These results suggest that the thermal decomposition of these classes of organic compounds may not be significant. However, we agree with the reviewer that some potential reaction products (e.g. oligomers and peroxides) could be thermally decomposed in our experiments.

We agree with the reviewer about the concerns about the thermal decomposition and quantification of the reaction products in our particle-DART technique. Since the

ionization efficiency of erythritol and its reaction products are currently not well understood, we thus do not attempt to quantify their concentrations.

We chose this detected range because this gave the best overall performance when we optimized our systems. Thus, small ions which were less $m/z$ 70 were not measured in our work. In this work, we primarily focus on the measurement of reaction products remained in the particle phase after oxidation. However, we agree with the reviewer that it would be important to measure these small ions, which are likely volatile fragmentation products present in the gas phase. The measurement of gas-phase products, together with the characterization of particle-phase products would provide better insight into heterogeneous reaction pathways.

**Comment #8:**
*Page 10, lines 13-14: Here again the statement is not correct. The authors have to be quantitative in order to make such a statement. For example, did the authors try to generate SOA with one single compound (e.g., carboxylic, polyol,...) and determine if the mass measured with the DART corresponds to the mass measure with an SMPS?*

**Author Response:**
On page 10, Line 21, we calculated the effective OH uptake coefficient for erythritol particles and erythritol–AS particles. We have not tried to generate SOA for the OH oxidation in this work.

As raised by the reviewer in his/her earlier comment, we do not attempt to quantify the particle mass measured with the DART and compare the results with that measured with an SMPS. One reason is that the ionization efficiencies of the reaction products are not known and not all reaction products were detected by the DART. While we cannot quantify the concentration of the reaction products, we can quantify the OH decay of the erythritol upon oxidation using their normalized signal before and after oxidation (Eqn.4 in the manuscript). This method does not require to know the mass concentration of the species.

**Comment #9:**
*Page 10, line 17: What would be the expected fragment ions? Not all ions were identified in Fig 1 and some fragments ions were not present before the reaction. Please clarify.*

**Author Response:**
Thanks for the comments. In terms of the expected fragment ion, we could not find the statement on Page 10, line 17 in the original manuscript. We think the reviewer may refer to Page 9, line 20: "but there was no indication of fragment ions expected from…". As mentioned in our response for **comment #7**, in the literature, the thermal decomposition of alcohols and carboxylic acids has not been observed and reported using particle-DART technique (Nah et al., 2013). However, we acknowledge that at high temperature organic peroxides may break down with the cleavage of O-O bond, and oligomers may also thermally decompose during the analysis to yield smaller fragmentation products (Mukundan and Kishore, 1990). The fragment ions could be difficult to identify depending on their chemical structures. Further investigation on the

formation of these products during heterogeneous OH oxidation of organic compounds is desirable. We have revised our statement in the manuscript.

Page 10, Line 2: "However, we would like to note that some possible reaction products (e.g. organic peroxides and oligomers) could be formed from reactions between peroxy radicals (Stark et al., 2017). We do not rule out the formation of these products upon OH oxidation of erythritol as they may undergo thermal decomposition at high temperature with the cleavage of O-O bond (Mukundan and Kishore, 1990). Further investigation on the formation of organic peroxides and oligomers upon heterogeneous OH oxidation of organic compounds is desirable."

We do not attempt to identify all ions in particle-DART mass spectra. One reason is that these ions were minor peaks with their relative abundances less than 2% (We could not rule out that their concentrations could be high since their chemical identities and ionization efficiencies are not known). Details about these ions are summarized in following table. We would like to acknowledge that $C_3H_3O_4^-$ ion ($m/z$ =103) was detected after oxidation at IOR = 0.0 but have been observed before oxidation at other IORs. Similarly, for $C_4H_3O_4^-$ ions ($m/z$ =115), it has been detected before oxidation at IOR = 1.0 and 5.0. We would also like to note that a minor ion at $m/z$ = 133 ($C_4H_5O_4^-$) was only presented after oxidation. Based on the suggested chemical formula, it may be a second-generation reaction product. To avoid confusion and overstatement, these three minor peaks were not discussed in this work.

| $m/z$ | Chemical formula | Relative abundance |
|---|---|---|
| New ion after oxidation at IOR = 0.0 | | |
| 103 | $C_3H_3O_4^-$ | 1.97 % |
| 115 | $C_4H_3O_4^-$ | 1.87 % |
| 133 | $C_4H_5O_5^-$ | 1.68 % |
| New ion after oxidation at IOR = 0.5 | | |
| 115 | $C_4H_3O_4^-$ | 1.86 % |
| 133 | $C_4H_5O_5^-$ | 1.33 % |
| New ion after oxidation at IOR = 1.0 | | |
| 133 | $C_4H_5O_5^-$ | 1.01 % |
| New ion after oxidation at IOR = 5.0 | | |
| 133 | $C_4H_5O_5^-$ | 0.51 % |

**Comment #10:**
*Page 10, Fig 2: Please explain the meaning of the error bars. Is it from different experiments?*

**Author Response:**
In Figure 2. (a), the x-error bar represents the calculated error in the OH exposure. The OH exposure, defined as the products of gas-phase OH concentration, [OH], and the particle residence time, t, was determined by measuring the decay of the hexane (Smith et al., 2009):

$$OH\ exposure = -\frac{ln([Hex]/[Hex]_0)}{k_{Hex}} = \int_0^t [OH]dt \ \text{(Eqn.1)}$$

where *[Hex]* is the hexane concentration leaving the reactor after oxidation, *[Hex]$_0$* is the initial hexane concentration before oxidation, and $k_{Hex}$ is the second-order rate constant of the gas-phase OH−hexane reaction).

Based on Eqn.1 and the error propagation rule, the uncertainty for OH exposure, $\sigma_{OH\ exp}$ ,was derived from Eqn.2:

$$\sigma_{exp} = 0.005\ (OH\ exposure)\sqrt{\left(16 + \frac{2}{(OH\ exposure \times k_{Hex})^2}\right)}\ (Eqn.2)$$

where 0.005 is the precision of Hex measurement (0.5 % of the reading). The y-error is the error in parent decay index ($\frac{I}{I_0}$), where *I* is the signal intensity of erythritol at a given OH exposure, *I$_0$* is the signal intensity before oxidation. The y-error was determined from the following equation when the uncertainty of signal intensity was assigned to be 0.1 %:

$$\sigma_{\frac{I}{I_0}} = \frac{I}{I_0} \times 0.1 \times \sqrt{2}\ (Eqn.3)$$

In Figure 2. (b), the y-error represents the uncertainty of derived effective OH uptake coefficient.

**Comment #11:**
*Page 17, line 15: Is it possible to estimate any branching ratio? Is the DART technique more sensitive/selective for certain types of compounds?*

**Author Response:**
Thanks for the suggestion. Since the ionization efficiencies of erythritol and reaction products are not known, the relative abundance of the products after oxidation cannot be well quantified. Furthermore, we are not able to detect all reaction products in both gas- and particle-phases. We thus do not attempt to estimate the branching ratios.

We agree with the reviewer that it is important to know whether the DART technique is more sensitive/selective for certain types of compounds. In the literature, the sensitivity and detection limits for some organic compounds has been reported to be dependent on the structure and functionality of investigated compounds (Cody et al., 2005; Nah et al., 2013; Chan et al., 2014; Chim et al., 2017a). For instance, Chan et al. (2014) have measured the ionization efficiency of oxalic acid, malonic acid, oxosuccinic acid, and tartaric acid relative to succinic acid and found these values can vary from 0.5 to 4.59 due to the change in carbon number ($C_2$ to $C_4$) and functional group (alcohol or ketone). To date, there are only limited classes of organic compounds were investigated. It remains largely unclear whether the particle-DART technique is more sensitive/selective for certain types of compounds. We would also like to note that the experimental configuration (Chan et al., 2013) and other factors (e.g. types of mass spectrometers) may have significant effects on the responses of organic compounds in the mass spectra (Nah et al., 2013). Cautions should be taken when we compare the results among different studies with various experimental configurations and conditions.

**Comment #12:**
*page 5, line 5. Should molecule cm$^{-3}$ s be molecule cm$^{-3}$ s$^{-1}$*

**Author Response:**
The reviewer may refer to the unit of OH exposure. The OH exposure, defined as the products of gas-phase OH concentration, *[OH]*, and the particle residence time, *t*, was determined by measuring the decay of the hexane (Smith et al., 2009):

$$OH\ exposure = -\frac{ln([Hex]/[Hex]_0)}{k_{Hex}} = \int_0^t [OH]dt \quad (Eqn.1)$$

where *[Hex]* is the hexane concentration leaving the reactor after oxidation, *[Hex]₀* is the initial hexane concentration before oxidation, and $k_{Hex}$ is the second-order rate constant of the gas-phase OH−hexane reaction ($5.2 \times 10^{-12}$ cm$^3$ molecule$^{-1}$ s$^{-1}$). Hence, the unit would be molecule cm$^{-3}$ s.

**Comment #13:**
*page 6, line 7. It should be Orbitrap*

**Author Response:**
We have revised the sentence.

Page 6, Line 1: "The gas-phase species leaving the heater were introduced into an atmospheric pressure ionization region, a narrow open space between the DART ionization source (IonSense: DART SVP), and the inlet orifice of the high-resolution mass spectrometer (Thermo Fisher, Q Exactive Orbritrap) for ionization and detection (Nah et al., 2013; Chan et al., 2013; 2014)."